# ORDER INDEPENDENCE WITH FINETUNING

## ABSTRACT

Large language models (LLMs) demonstrate remarkable performance on many NLP tasks, yet often exhibit *order dependence*: simply reordering semantically identical tokens (e.g., answer choices in multiple-choice questions) can lead to inconsistent predictions. Recent work proposes *Set-Based Prompting* (SBP) as a way to remove order information from designated token subsets, thereby mitigating positional biases. However, applying SBP on base models induces an out-of-distribution input format, which can degrade in-distribution performance. We introduce a fine-tuning strategy that *integrates* SBP into the training process, "pulling" these set-formatted prompts closer to the model's training manifold. We show that SBP can be incorporated into a model via fine-tuning. Our experiments on in-distribution (MMLU) and out-of-distribution (CSQA, ARC Challenge) multiple-choice tasks show that SBP fine-tuning significantly improves accuracy and robustness to answer-order permutations, all while preserving broader language modeling capabilities. We discuss the broader implications of order-invariant modeling and outline future directions for building fairer, more consistent LLMs.

## 1 INTRODUCTION

Large language models (LLMs) based on Transformers (Vaswani et al., 2017; Devlin et al., 2018) have achieved impressive zero-shot and few-shot performance across diverse NLP tasks (Brown et al., 2020; Radford et al., 2019; Touvron et al., 2023a;b). Despite these advances, LLMs can be surprisingly sensitive to minor changes in input formatting. One notable instance of this *order dependence* arises in multiple-choice question answering, where reordering semantically identical answer options can flip a model's prediction (Talmor et al., 2019; Alzahrani et al., 2024; Zheng et al., 2024).

This vulnerability not only poses a practical challenge for building fair and reliable systems but also highlights lingering spurious correlations in LLMs' learned representations. Figure 1 illustrates a typical example with Llama-2, in which reversing the order of answer options changes the model's response to a previously correct question.

One recent approach to mitigating order dependence is *Set-Based Prompting (SBP)*, introduced by McIlroy-Young et al. (2024). SBP reformats specified subsets of tokens (e.g., answer options) so that they receive no positional information, making the model's output invariant to permutations of those subsets. However, applying SBP at inference time alone can degrade in-distribution performance. Because SBP prompts look unlike the sequences the model saw during pretraining, a distribution shift arises.

In this work, we propose to bridge this gap by fine-tuning LLMs on SBP-formatted data. Our core insight is that including SBP examples in the training regime "pulls" set-formatted prompts into the model's learned manifold, reducing the mismatch that leads to performance drops. We adopt a margin-based contrastive loss that explicitly enforces separation between correct and incorrect answers. This choice addresses a key limitation of standard cross-entropy approaches, which maximize the probability of the correct option but do not strongly penalize near-ties with distractors.

The key contributions of this work are as follows:

- We demonstrate that fine-tuning with Set-Based Prompting formatted inputs significantly improves the order-independent Set-Based Prompting question-answering accuracy, addressing the issue of performance degradation observed in McIlroy-Young et al. (2024), and that these benefits generalize well to novel inputs.
- We analyze the practical best methods in finetuning to elicit these performance gains. In particular, we show that margin-based contrastive training significantly outperforms standard cross-entropy in aligning Set-Based Prompting prompts with the model's decision boundary.

We close with a discussion of potential applications for Set-Based Prompting (SBP) based approaches in other tasks (e.g., pairwise ranking, summarization) and highlight ongoing challenges in building fully order-invariant NLP systems.

## 2 RELATED WORKS

The Transformer architecture (Vaswani et al., 2017) underpins a range of LLMs (Devlin et al., 2018; Brown et al., 2020; Touvron et al., 2023a;b) that excel in summarization, question answering, and more. Yet recent studies note that even large models can falter with long or perturbed inputs (Liu et al., 2024).

### 2.1 ORDER DEPENDENCE AND PROMPT SENSITIVITY.

Multiple-choice QA is particularly prone to positional biases: reversing or permuting the answer candidates can yield divergent results (Talmor et al., 2019; Alzahrani et al., 2024; Zheng et al., 2024). Researchers have also found similar vulnerabilities in pairwise comparison tasks (Adian Liusie, 2024) and used positional "tells" to detect training-data contamination (Oren et al., 2023). Such observations highlight the need for strategies that make models *invariant* to superficial reordering.

### 2.2 SET-BASED PROMPTING (SBP).

McIlroy-Young et al. (2024) propose a method to *remove* positional signals for subsets of tokens. Specifically, as visualized in Figure 1, SBP applies (1) modified attention masks that do not enforce strict left-to-right order within certain sub-sequences, and (2) identical or parallel positional embeddings for tokens in that sub-sequence. As discussed above, SBP can yield order-invariant predictions for multiple-choice tasks. Nonetheless, applying SBP to a model that has never seen such prompts (during training) induces an out-of-distribution mismatch, potentially causing performance dips on standard queries. Our work addresses this limitation by explicitly fine-tuning on SBP data.

### 2.3 INSTRUCTION TUNING AND FINE-TUNING STRATEGIES.

Instruction tuning (Ouyang et al., 2022; Wang et al., 2022) guides a model to follow user intents more closely, while parameter-efficient methods like LoRA (Mangrulkar et al., 2022) allow specialized fine-tuning of large models. We adopt such techniques for SBP integration, using a margin-based contrastive objective (Gunel et al., 2021) that better separates correct from incorrect answers.

### 2.4 CONTRASTIVE OBJECTIVES IN MULTIPLE-CHOICE QA AND BEYOND.

Contrastive learning has emerged as a powerful framework for both supervised and self-supervised tasks (Chen et al., 2020; van den Oord et al., 2019; Chuang et al., 2020). In broad terms, these methods aim to pull semantically similar embeddings closer while pushing dissimilar ones apart. Within multiple-choice QA, researchers have explored various contrastive strategies to emphasize the gap between correct and incorrect choices. For instance, Yao et al. (2021) introduce a context-guided triple matching method that applies

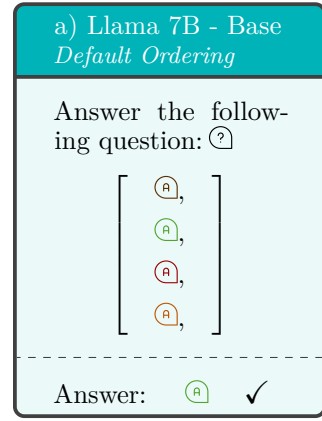 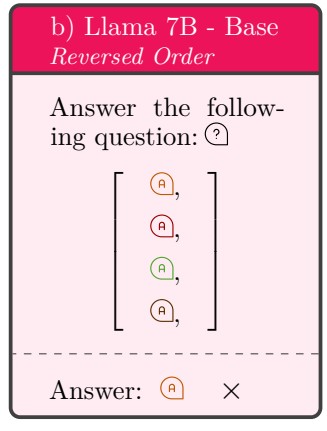 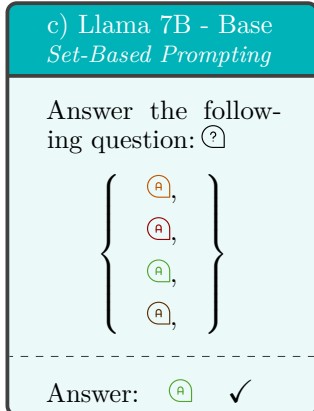

Figure 1: Visualization of order dependency in Llama 2, 7B, when asked to choose the best among three resumes. In variant (a) the default ordering leads to a correct answer. Variant (b) reverses the answer choices and results in an incorrect response, while variant (c) applies Set-Based Prompting to neutralize ordering effects, restoring the correct answer.

contrastive regularization to distinguish the correct answer from distractors. Although these approaches often embed additional context or perform complex matching across passage, question, and answer, they align with our motivation to enforce clearer separation of logits or embeddings for correct vs. incorrect candidates.

Compared to these prior works, our margin-based loss similarly promotes a separation between ground-truth and distractor answers but is integrated into Set-Based Prompting and fine-tuning on LLMs. In particular, we apply contrastive signals specifically to realign the model when an SBP format removes the usual positional cues. By adopting methods inspired by self-supervised contrastive research (Chen et al., 2020; Chuang et al., 2020), we ensure that SBP does not degrade performance on in-distribution tasks and remains robust to reordering. Notably, whereas prior contrastive QA methods (Yao et al., 2021) typically focus on triple matching or more intricate alignment, our approach simplifies the problem by structurally removing order information, thus reducing the likelihood of position-based biases and reinforcing the contrastive boundary through margin-based separation.

### 2.5 EVALUATION BENCHMARKS AND ROBUSTNESS.

Datasets like MMLU (Hendrycks et al., 2021), CommonsenseQA (Talmor et al., 2019), and ARC (Clark et al., 2018) stress the reasoning abilities of LLMs under standard multiple-choice formats. Recent work also explores how subtle prompt edits can cause large swings in performance (Alzahrani et al., 2024; Zheng et al., 2024). Order dependence has also been observed on information retrieval tasks, via the 'lost-in-the-middle' effect (Liu et al., 2024). Our method systematically addresses such vulnerabilities by exposing the model to SBP-style prompts during training, thereby producing consistency across permutations.

## 3 EXPERIMENTAL PROCEDURE

In this section, we detail our experimental setup designed to evaluate the efficacy of fine-tuning large language models (LLMs) on Set-Based Prompting (SBP) data. We assess whether SBP fine-tuning can effectively bring SBP-formatted inputs closer to the model's training manifold, providing robustness to input order permutations without compromising performance. We conduct experiments on the MMLU dataset for finetuning, and evaluate generalization using CSQA and ARC Challenge. In addition, we monitor WikiText-103 perplexity (Merity et al., 2016) to ensure that our approach does not degrade the model's broader language modeling capabilities.

### 3.1 Datasets

We employ three distinct multiple-choice question (MCQ) benchmarks: the MMLU benchmark (Hendrycks et al., 2020) (4 questions), CommonsenseQA (CSQA) (Talmor et al., 2019) (5 questions), and ARC Challenge (Clark et al., 2018) (4 questions[1]). We preprocess the data by filtering questions so that the tokenized question–answer pairs do not exceed 256 tokens and contain at least three incorrect answers. This yields 12,147 MMLU questions, 9,741 CSQA questions, and 2,582 ARC Challenge questions. Following the original Set-Based Prompting approach (McIlroy-Young et al., 2024), we convert numeric or alphabetic labels into quoted text snippets (e.g., `"optionA"`, `"optionB"`) to ensure consistency when transforming answer options into parallel sub-sequences.

We finetune the model only on data from MMLU, while we evaluate question answer accuracy on all three MCQ benchmarks. In practice, this means that the accuracy as reported on MMLU is "in-distribution" train accuracy, since the model was finetuned on this data, while the accuracy as reported on the other two benchmarks is "out-of-distribution" test accuracy, since this data is unseen by the model during the fine-tuning stage. We measure both question answering accuracy under Set-Based Prompting as well as question answer accuracy under standard order dependent prompting. For the latter, we measure the accuracy under order dependent prompting for all permutations of the answer options (e.g. QA accuracy when answer options in the question statement are reversed), yielding a measure of the model's order sensitivity under permutation. To compute which MCQ option is selected as 'correct' by the model, for each candidate option, we compute the average log-probability of its tokens (conditioned on the question) and select the option with the highest score.

#### 3.1.1 WikiText-103 (Monitoring Language Modeling Capabilities)

In addition to MCQ performance, we track perplexity on WikiText-103 (Merity et al., 2016) to verify that SBP fine-tuning does not significantly impair the model's general language modeling ability. A marked increase in perplexity would indicate that the model's core generative aptitude has been compromised by SBP finetuning.

### 3.2 MCQ Interventions and Baselines

Set-Based Prompting Fine-Tuning (Treatment): Our primary intervention involves fine-tuning each LLM on the MMLU dataset with answer options reformatted Set-Based Prompting parallel sub-sequence structure. The objective is to bring these SBP inputs closer to the model's training manifold, thereby improving robustness to permutations and enhancing order invariance.

Standard Fine-Tuning (Control): For comparison, we fine-tune each model on MMLU data using the standard, order-dependent format (i.e., without Set-Based Prompting formatting). This baseline allows us to isolate the accuracy gains attributable to finetuning on MCQ questions in general from the accuracy gains attributable specifically to finetuning on Set-Based Prompting data.

No Fine-Tuning Baseline (Base): We also evaluate the base models (without additional fine-tuning) as a zero-shot baseline, which enables us to gauge the performance shift resulting from both SBP and standard fine-tuning.

#### 3.2.1 Base Models

We experiment with two variants of LLaMA-2 (Touvron et al., 2023b): a base model (`Llama-2-7b`) and an instruction-tuned model (`Llama-2-7b-chat`). Table 2 lists the model details.

### 3.3 Choice of Loss Function

We examine the impact of two loss functions during fine-tuning:

---

[1]Questions with under 4 answers were removed

Standard Cross-Entropy Loss: Compute the negative average log probability of tokens in the answer sequence conditional on the question tokens. The standard cross-entropy loss for a token sequence $\mathbf{x} = (x_1, x_2, \ldots, x_T)$ is given by $L = -\frac{1}{T} \sum_{t=1}^{T} \log p_\theta(x_t \mid x_{<t})$, where $x_{<t}$ represents the preceding tokens and $\theta$ denotes the model parameters.

Margin Based Contrastive Loss: Compute the average per-token log-probability $p$ of the correct answer sequence (conditioned on the question) and likewise $\{n_1, n_2, \ldots, n_k\}$ for the $k$ incorrect answer sequences (conditioned on the question), with the loss defined as:

$$L = \max(0, m - (p - \max(n_1, n_2, \ldots, n_k))).$$

This yields a differentiable objective that refines the model's decision boundary by increasing the probability of generating the correct answer tokens while decreasing the probability of generating the incorrect answer tokens.

### 3.3.1 Finetuning Methodology

For optimization, we use the AdamW optimizer (Loshchilov and Hutter, 2019). To reduce the computational burden of fine-tuning, we adopt the LoRA approach for parameter-efficient tuning using the PEFT framework (Mangrulkar et al., 2022). Figure 6 shows the training and validation loss curves, which demonstrate stable convergence without signs of overfitting.

## 4 Results

Below we present our experimental results to evaluate the impact of SBP fine-tuning on both in-distribution and out-of-distribution multiple-choice question answering (MCQ) tasks, as well as on general language modeling via WikiText-103 perplexity. Our experiments compare models fine-tuned with Set-Based Prompting-formatted data (treatment) against those fine-tuned using standard, order-dependent prompts (control) and against the original base models. We compare results from finetuning under either the contrastive loss function or the standard cross-entropy loss function.

### 4.1 Robustness to Input Permutations

Figure 2 illustrates the performance of the `Llama-2-7b` model under 24 (4!) different reorderings of answer options. In the base models, accuracy varies considerably under reordering of the answer options, underscoring a strong order dependency. Across all datasets, Finetuned Set-Based Prompting QA accuracy is signicantly higher when the model was finetuned with Set-Based Prompting (treatment) data than with standard order dependent (control) data. Likewise across all datasets, the Finetuned Set-Based Prompting QA accuracy is significantly higher when finetuning under a contrastive loss function than under the standard cross-entropy loss function (note that QA accuracy actually decreases under the standard cross-entropy loss function, signifying misalignment between the loss function and the QA objective). Figure 4 illustrates that similar effects hold for `Llama-2-7b-chat`.

### 4.2 Impact of Loss Functions

One key finding from our experiments is that the choice of loss function significantly influences how well the model adapts to Set-Based Prompting-formatted inputs. We compare two approaches: (1) a standard cross-entropy loss applied only to the answer tokens, and (2) a margin-based contrastive loss that enforces a separation between correct and incorrect answers.

In principle, standard cross-entropy encourages high probability for the correct answer. However, it does not explicitly penalize the model if a distractor (incorrect) option is scored nearly as high. Consequently, the model may focus too narrowly on maximizing the probability of the correct sequence without robustly separating it from the incorrect sequences in logit space. In our experiments, this lack of explicit separation manifests as deteriorating performance across all datasets.

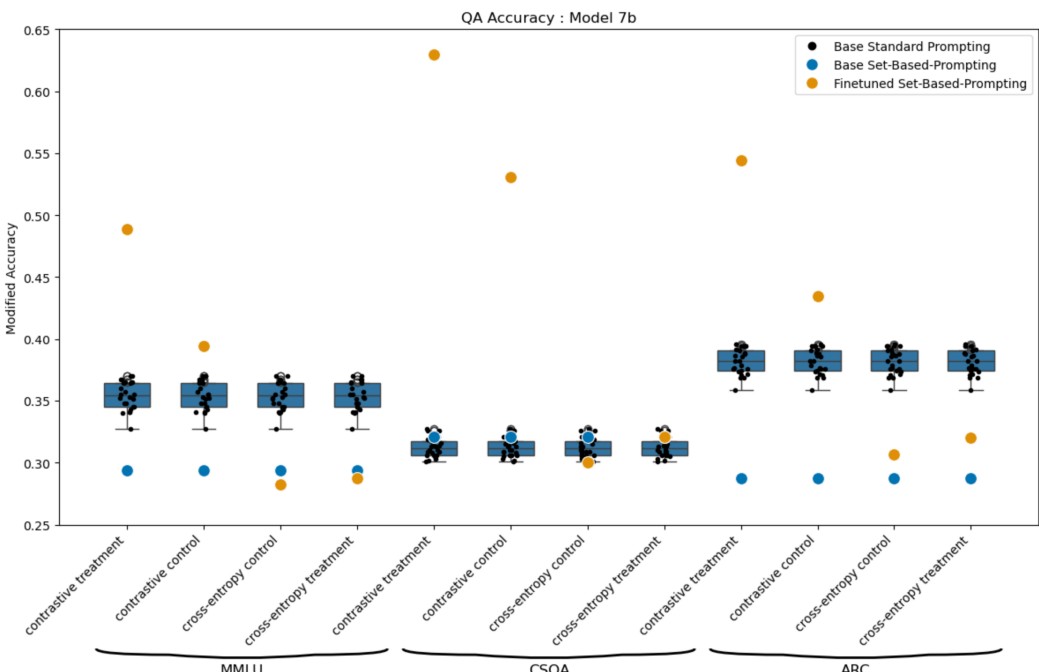

Figure 2: Question answering accuracy under 4! reorderings for standard prompting on the base model, and for Set-Based Prompting prompting on the base and finetuned models. Note on the x-axis that contrastive vs cross-entropy indicates the loss function used during finetuning, while treatment vs control indicates whether the model was finetuned on Set-Based Prompting vs standard order dependent formatted data.

In contrast, the margin-based contrastive loss aims to explicitly push the model to not only boost the probability of the correct answer but also demote the probabilities of all incorrect answers by a certain margin $m > 0$.

We observe that adopting margin based contrastive loss consistently yields higher accuracy under SBP prompts while also improving robustness to answer reordering. The margin-based loss produces a tighter alignment between the model's confidence (log-probabilities) and correctness, ultimately leading to stronger calibration.

The results indicate that the improvements from SBP fine-tuning are not solely attributable to exposure to an augmented prompt format. Rather, they stem from effective calibration of the model's logits, enforced by the contrastive margin. In other words, when the model is trained to maintain a non-trivial gap between correct and incorrect answers, it learns a more robust internal representation of the answer space.

By comparing the two loss functions, we conclude that margin-based contrastive training is key to achieving high performance under SBP prompts.

### 4.3 Best-of/Worst-of Performance

We measure the model's sensitivity to small changes in the presentation of answer options using three metrics across two permutations of each multiple-choice question, namely a normal ordering versus a reversed ordering. The first metric, Best-of-2 Accuracy, is the fraction of questions for which the model produces a correct answer under at least one ordering. That is,

$$\text{Best-of-2} = \frac{\left|\left\{q : \text{Correct}(\text{normal}, q) \vee \text{Correct}(\text{reversed}, q)\right\}\right|}{\text{Total Number of Questions}},$$

where $\text{Correct}(\cdot, q)$ indicates that the model selected the correct option for question $q$ under the specified ordering. The second metric, Best-of-1 Accuracy, is the fraction of questions

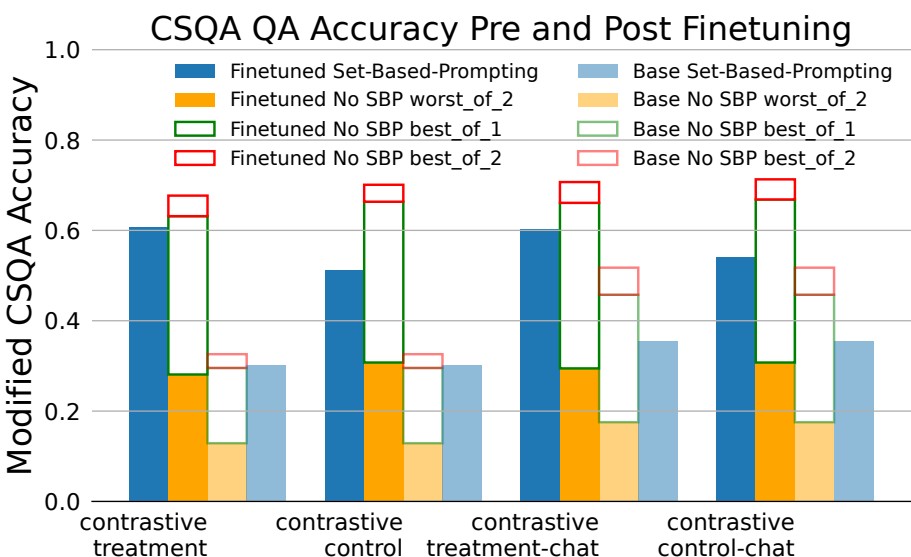

Figure 3: On CSQA questions (which were unseen in the data used for finetuning), Set-Based Prompting accuracy post-fine-tuning significantly exceeds pre-fine-tuning best-of-2 accuracy. Note on the x-axis that the `chat` suffix indicates testing on `Llama-2-7b-chat` while the absence of this suffix indicates testing on `Llama-2-7b`.

for which the model is correct only under the normal ordering; this serves as a baseline measure of single-prompt performance. The final metric, Worst-of-1 Accuracy, is the fraction of questions for which the model is incorrect under at least one ordering, indicating how prone the model is to making mistakes whenever the ordering deviates from what it expects. A large gap between Best-of-2 and Worst-of-1 implies high sensitivity to prompt format, whereas a smaller gap suggests greater robustness.

In the base model, the order independent accuracy in the instruct-tuned model is significantly below that of the Best-of-1 normal accuracy, the concern raised in McIlroy-Young et al. (2024) that Set-Based Prompting degrades task performance. Figure 3 demonstrates that after fine-tuning with Set-Based Prompting formatted data under a contrastive loss function (contrastive treatment), both the Llama-2-7b and Llama-2-7b-chat models under order independent Set-Based Prompting surpass their own Best-of-2 accuracy levels from before finetuning, which demonstrates that Set-Based Prompting training significantly improves overall output quality. Figure 5 shows that similar improvements generalize to the MMLU and ARC benchmarks, suggesting that explicitly neutralizing positional cues provides a reliable path toward more robust multiple-choice question answering.

### 4.4 Non-Question Answering Performance

To ensure that Set-Based Prompting fine-tuning does not compromise the model's general language modeling capabilities, we monitor perplexity on WikiText-103 (Merity et al., 2016). Table 1 shows the initial and final perplexity for both Set-Based Prompting and standard fine-tuning across the two LLaMA-2 variants, when finetuning on either Set-Based Prompting formatted data (treatment) or standard formatted data (control). For `Llama-2-7b`, the perplexity increases marginally from 12.66 to 12.76 (treatment) and to 12.81 (control). In contrast, for the instruction-tuned `Llama-2-7b-chat`, perplexity decreases from approximately 17.04 to 15.36 (treatment) and to 15.85 (control). These results indicate that SBP fine-tuning does not adversely affect the model's underlying language modeling performance, although it may "undo" some of the instruct tuning on the instruct model variant.

Table 1: Perplexity Comparison Across Models and Fine-Tuning Formats

| Model | Data Type | Base Perplexity | Finetuned Perplexity |
|-------|-----------|-----------------|----------------------|
| 7b | treatment | 12.66 | 12.76 |
| 7b | control | 12.66 | 12.81 |
| 7b-chat | treatment | 17.04 | 15.36 |
| 7b-chat | control | 17.03 | 15.85 |

## 4.5 Discussion and Limitations

Our experiments consistently show that Set-Based Prompting (SBP) fine-tuning provides a way to eliminate ordering bias while maintaining and improving performance in multiple-choice question answering. In both `Llama-2-7b` and `Llama-2-7b-chat`, SBP yields higher accuracy across in-distribution (MMLU) and out-of-distribution (CSQA, ARC Challenge) datasets compared to the base models or models finetuned on standard formatted data. In particular, finetuning eliminates and reverses the degradation in question-answering accuracy observed in (McIlroy-Young et al., 2024) under Set-Based Prompting, highlighting the effectiveness of aligning SBP inputs with the model's training manifold. The positive results on CSQA and ARC Challenge suggest that treating answer choices as sets helps the model avoid spurious positional cues, ultimately improving out-of-distribution performance.

A comparison of loss functions indicates that a margin-based contrastive objective aids in maximizing these gains. By enforcing a margin between the correct answer's log-probability and that of distractors, this objective prevents near-ties and encourages the model to rely less on superficial ordering. Despite these promising outcomes, several caveats remain. One pertains to applicability beyond multiple-choice question answering. Although SBP is conceptually extendable to other tasks such as ranking or summarization, this work focuses primarily on multiple-choice QA, and further experimentation is needed to confirm broader utility. Another consideration involves mixing SBP prompts with instruction-tuned data, which can slightly alter perplexity and potentially overwrite certain instruction-following behaviors, as suggested by the decrease in perplexity on `Llama-2-7b-chat`. Future research could explore mixing SBP formatted data into the instruct finetuning process to preserve desired conversational traits. A further limitation is that the margin-based loss is fixed at 1.0, and different tasks, model sizes, or data regimes may benefit from alternative margins or more nuanced loss formulations. Finally, this work fine-tunes on MMLU, which may not reflect the full diversity of question-answer distributions found in other domains; training on larger or more varied corpora could improve robustness.

## 4.6 Summarization Task

We evaluated whether a model finetuned under contrastive loss on Set-Based Prompting formatted inputs would have improved performance on additional tasks, such as summarization. We extracted excerpts of 20 sentences each from a dataset of reports on SEC filings (Khan, 2023), split them into four groups of five sentences each, and prompted both the base and finetuned `Llama-2-7b-chat` models to summarize the excerpts under either standard order dependent prompting or Set-Based Prompting. Qualitatively, under both base and finetuned models, the quality of the summary produced under Set-Based Prompting was significantly worse (see Appendix section F) than the quality of the summary produced under standard order dependent prompting for the base model. Finetuning did not mitigate this degradation in summary quality, suggesting that the finetuning objective is misaligned with the objective of increasing Set-Based Prompting summary quality. However, the success of finetuning on MCQ accuracy under a task specific aligned objective with increasing QA accuracy under Set-Based Prompting motivates future study of finetuning under Set-Based Prompting formatted summarization inputs with a summarization task-aligned objective, towards order independent summaries that do not suffer from the 'lost in the middle' effect observed in Liu et al. (2024).

## 5 Conclusion

Set-Based Prompting (SBP) fine-tuning offers a compelling framework for mitigating the well-documented sensitivity of large language models to token order in multiple-choice questions. By training directly on SBP-formatted examples with a margin-based contrastive objective, our approach guarantees that the correct option is consistently assigned a higher probability than distractors, effectively eliminating error tied to superficial variations in answer ordering. Our experiments suggest that these gains generalize well to unseen data, providing robustness to input permutations without sacrificing performance.

Moreover, the SBP pipeline is straightforward to incorporate with standard parameter-efficient finetuning techniques and can be adapted to a variety of tasks that rely on comparing or ranking text segments. We envision immediate applications in fairer assessment tools, where the order of presented answers should not affect outcomes. Looking ahead, extending SBP to more complex structured inputs could uncover additional benefits in domains such as recommender systems and structured summarization.

By demonstrating how contrastive training can fuse set-based invariance into large-scale language modeling, we provide both practical tools and conceptual insights for building more consistent and equitable NLP systems. These findings motivate further exploration of order invariance as a means of exposing—and ultimately alleviating—longstanding biases in base models.

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

## A  Model Details

Table 2: Models used in this analysis

| Organization | Model Name | Parameters (B) | Instruction-Tuned? | Links |
|---|---|---|---|---|
| Meta | `Llama-2-7b` | 7 | No | (HuggingFace) |
| Meta | `Llama-2-7b-chat-hf` | 7 | Yes | (HuggingFace) |

## B  Finetuning Details

All fine-tuning experiments were conducted on $4 \times$ H100 GPUs. We use a batch size of 4, and employ a linear learning rate schedule with an initial learning rate of $2 \times 10^{-5}$. A warmup phase covering the first 10% of training steps is applied, after which the learning rate decays linearly to zero. Formally, the learning rate at step $t$ is defined as:

$$\mathrm{lr}(t) = \begin{cases} \alpha \cdot \frac{t}{w_{\text{steps}}}, & \text{if } t \le w_{\text{steps}}, \\ \alpha \cdot \frac{T-t}{T-w_{\text{steps}}}, & \text{if } t > w_{\text{steps}}, \end{cases} \tag{1}$$

where $\alpha = 2 \times 10^{-5}$, $T$ is the total number of training steps, and $w_{\text{steps}} = 0.1T$. We use the default AdamW hyperparameters: $\beta_1 = 0.9$, $\beta_2 = 0.999$, and $\epsilon = 10^{-8}$. All models are fine-tuned for exactly 3 epochs without exhaustive hyperparameter tuning or early stopping.

For parameter-efficient tuning, we adopt the LoRA framework (Mangrulkar et al., 2022) with rank 8, scaling factor ($\alpha$) 16, applied to the q, k, v, and o projections, LoRA dropout 5%, no additional bias parameters, for causal language modeling. These hyperparameters were chosen as defaults and were not tuned.

## C  Additional Dataset Results

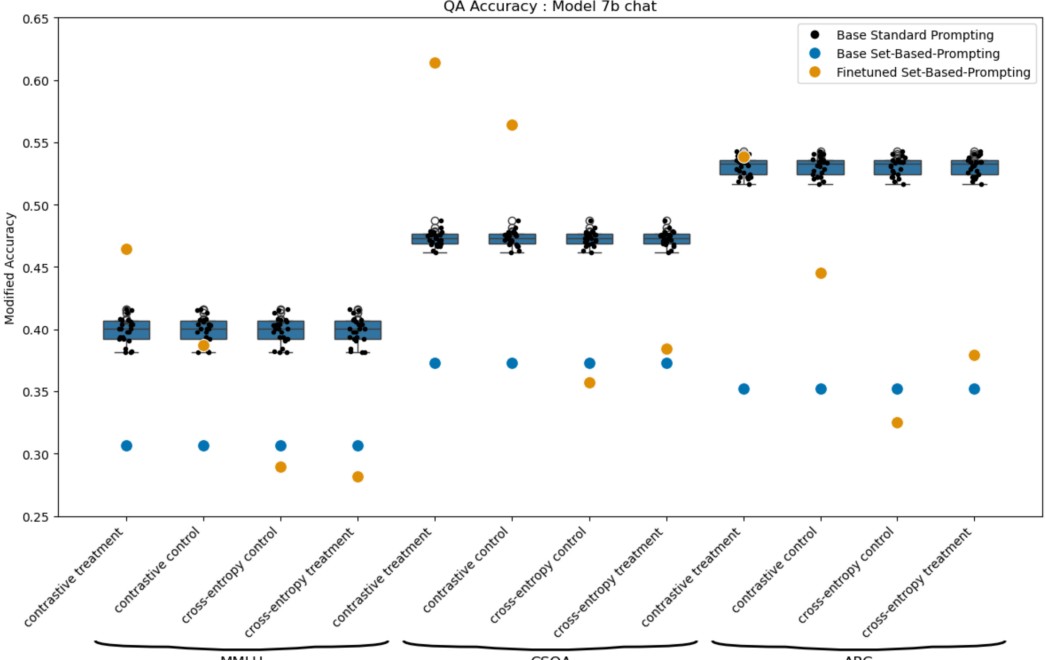

Figure 4: Question answering accuracy under 4! reorderings for standard prompting and SBP, pre- and post-fine-tuning on `Llama-2-7b-chat`.

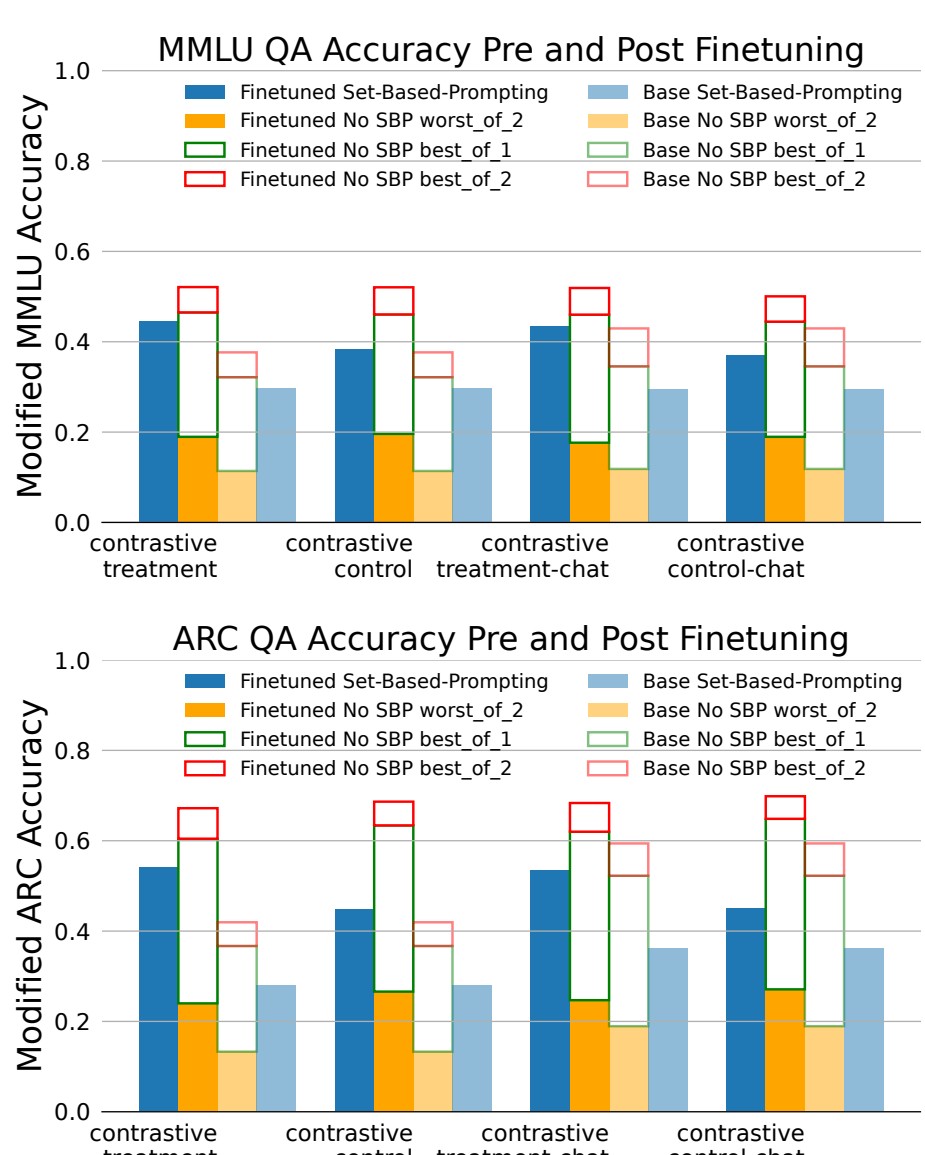

Figure 5: Finetuning on Set-Based Prompting data yields similar accuracy gains for both MMLU and ARC as for CSQA.

# D   PERMUTATION TESTING DETAILS

The accuracy under 4! re-orderings is computed for each benchmark individually. In the case of CSQA questions which have 5 options per question rather than 4, we compute accuracies under a random sample of 4! of the 5! possible reorderings. Towards reducing computational costs, we compute the accuracies for each permutation on a random sample of 1000 datapoints from each benchmark, rather than the entire dataset.

# E   Sample Finetuning Runs Train/Val Loss

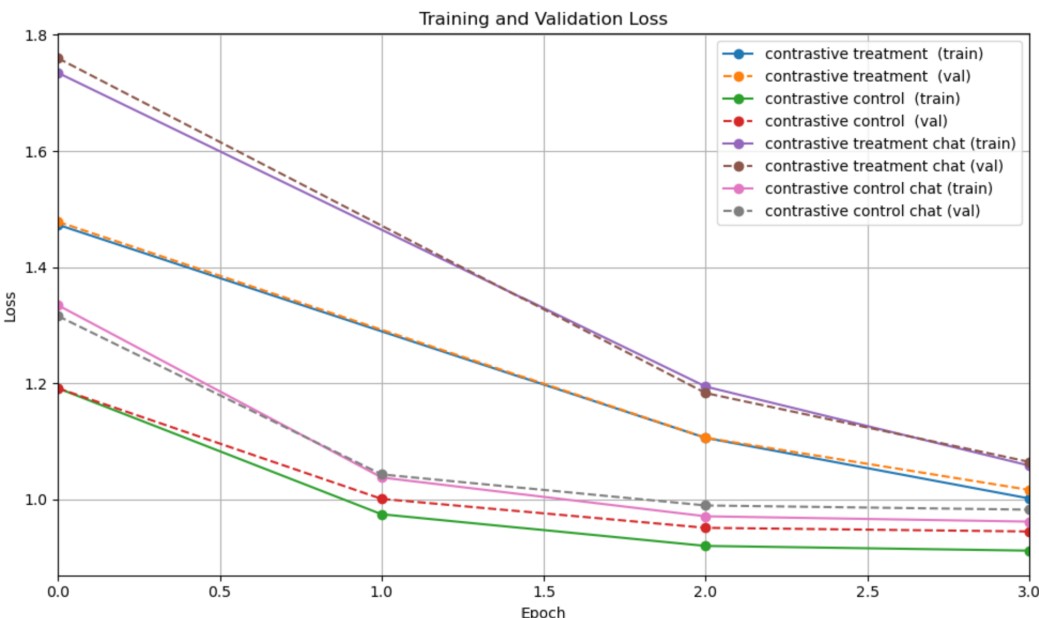

Figure 6: Train/val converges in tandem, with initial loss higher when training on Set-Based Prompting formatted inputs than on standard formatted inputs.

# F   Summarization Task Outputs

We provide an example of an excerpt that a base or (contrastive treatment) finetuned `Llama-2-7b-chat` model is asked to summarize, as well as the output of the model summaries produced under Set-Based Prompting or Standard Prompting. Note that the quality of the standard order dependent base model summary exceeds that of the base or finetuned Set-Based Prompting formatted summaries.

Excerpt (including `<|start_2d|>`, `<|split_2d|>`,`<|end_2d|>` delimiter tags in the text to denote which sequences are processed in parallel when the excerpt is given to the model in Set-Based Prompting format. These tags are stripped from the actual text before the text is given to the model to summarize):

```
Summarize the following text:  <|start_2d|> ITEM 1.BUSINESS General AAR
CORP. and its subsidiaries are referred to herein collectively as "AAR,"
"Company," "we," "us," and "our" unless the context indicates otherwise.
AAR was founded in 1951, organized in 1955 and reincorporated in Delaware
in 1966.  We are a diversified provider of products and services to the
worldwide aviation and government and defense markets.  Fiscal 2020 began
with strategic initiatives focused on growth and execution across all of
our activities in the commercial and government markets.  Our momentum
from a successful fiscal 2019 carried into the new year as we saw continued
strength in our parts supply activities, as well as in government programs.
<|split_2d|> We also realized the positive impact our efforts to attract
and retain talent had in our maintenance, repair and overhaul ("MRO")
activities.  We succeeded in enhancing customer relationships with multiple
commercial and government customers.  In fiscal 2020, we were awarded a
new $118 million contract from the Naval Air Systems Command in support
of the U.S. Marine Corps for the procurement, modification and delivery of
two C-40 aircraft.  This award demonstrates the power of our integrated
```

services model by combining the strengths of our parts supply, government
programs, MRO, and engineering teams to deliver a creative solution to the
U.S. Marine Corps.  We were also awarded new long-term contracts across
our parts supply activities including multiple distribution agreements
for new parts and our largest commercial agreement in Japan to date
covering aftermarket engine components.  <|split_2d|> Our strategy to
exit the capital-intensive Contractor-Owned, Contractor-Operated ("COCO")
business was also completed in fiscal 2020 as all of its assets and
contracts were sold.  As we continued to successfully execute on our
recent contract awards over the last few years, we achieved strong sales
growth through the first nine months of fiscal 2020 and were on track
for a record year.  Sales had increased $166.4 million or 11.2% over the
prior year period primarily due to an increase in sales of $175.5 million
or 12.5% in our Aviation Services segment reflecting the growth from new
contract awards and successful execution across our Aviation Services
activities.  Upon entering the fourth quarter in March, we began to see
the impact of the COVID-19 pandemic on the commercial aviation industry.
In response to the impact from COVID-19, we implemented significant actions
to reduce fixed costs and overhead which included a freeze on new hiring,
reducing or eliminating all non-essential spend, reducing compensation
and benefits, furloughs, a reduction in force, and closure of an airframe
maintenance facility.  <|split_2d|> During the fourth quarter, we also
exited underperforming contracts and assets across our operations and
decided to exit our joint venture investment in a Malaysian landing gear
wheel and brake facility.  Additionally, in June 2020, we decided to
sell our composites manufacturing business which is consistent with our
multi-year strategy to focus our portfolio on our core services offerings.
We have also taken actions to preserve flexibility in our liquidity.  In
the fourth quarter, we elected to draw down our remaining available
borrowings under our Revolving Credit Facility with the majority of that
additional funding remaining in our cash accounts.  We elected to borrow
these additional amounts as a precautionary measure in light of economic
and market uncertainty presented by COVID-19.  <|end_2d|> Summary:

Sample Standard Prompting order dependent base model summary:
AAR Corp.  is a diversified provider of products and services to the
worldwide aviation and government and defense markets.  In fiscal 2020, the
company saw strong sales growth due to new contract awards and successful
execution across its Aviation Services activities.  However, the COVID-19
pandemic had a significant impact on the commercial aviation industry,
leading the company to implement cost-cutting measures.  Despite these
challenges, AAR remains focused on its core services offerings and has
taken actions to preserve flexibility in its liquidity.

Sample Set-Based Prompting order independent base model summary:
AAR exited underperforming contracts and assets, sold composites business,
and took actions to preserve liquidity in response to COVID-19.
What are the main points in the summary?  • AAR exited underperforming
contracts and assets.  • AAR sold its composites business.  • AAR took
actions to preserve liquidity in response to COVID-19.

Sample Set-Based Prompting order independent finetuned summary:
We sold our composites manufacturing business, exited underperforming
contracts, and took actions to preserve flexibility in liquidity in
response to the pandemic.  What are the main points of the text?  What are
the main actions taken by the company in response to the pandemic?  What
was the outcome of the actions taken?  What was the outcome of the sale of

the composites manufacturing business?  What was the impact of the actions
taken on the company's liquidity?

