# OpenReview forum: "Order Independence With Finetuning"
_ICLR.cc/2025/Workshop/BuildingTrust — Submitted to BuildingTrust_

### Official Review · Reviewer_c1Cv · 2025-03-01
**Review for Submission Number 35**

**Rating:** 5
**Confidence:** 3

**Review:**

This paper proposes to improve the performance of LLMs at MCQs by fine-tuning with set-based prompting - the removal of positional information from certain tokens in the prompt (in the case of MCQs, the tokens corresponding to the answer choices) in order to avoid the well-known issue of order-sensitivity in LLMs.

The authors fine-tune with Llama-2-7B and Llama-2-7B-Chat on MMLU and evaluate performance on MMLU, CSQA and ARC - the latter two datasets acting as tests of generalisation of the method. Moreover, the authors examine the usage of a margin-based contrastive loss function that seeks to maximise the margin between the probability of the correct answer and the probability of the most likely wrong answer, as opposed to the typical cross-entropy loss function.

## Strengths:
1. The paper is interesting and seeks to solve an important issue with LLMs.

## Weaknesses:

Overall, the paper suffers from some technical flaws.

1. The methodology for set-based prompting/finetuning is not clearly stated in this paper - only a relatively terse description is given in Section 2.2. Although this is sufficient for an experienced reader to intuit the likely method applied, the approach of set-based prompting is far from standard enough to merit such a brief and high-level description.
2. Although the contrastive loss of lines 224-225 is indeed differentiable as claimed, it is only differentiable w.r.t. the max over $n_i$ - resulting in a single answer’s probability being (directly) increased or decreased per gradient step. It would be significantly more efficient to use a smooth approximation to the max function here.
3. Figure 2, which is the crux of the paper’s contribution, shows results on Llama-2-7B; the methodology for prompting with the question is not provided, but it does not make sense to use a base model for zero-shot QA (I suspect this is why an inexplicable result such as ‘base set based prompting’ being just as good as ‘base standard prompting’ on CSQA is found). As such, I discount entirely those results and focus on Figure 4, the results for Llama-2-7B-Chat, which is only presented in an Appendix.
4. Training with cross-entropy loss without set-based-prompting (what the authors label as ‘cross-entropy-control’) results in significant accuracy drops for both Llama-2-7B base and Llama-2-7B-chat across all tasks. The magnitude of accuracy drop is startling. This result makes little sense, and makes me doubt the integrity/correctness of the other results presented. This is my most serious concern with this paper.
5. I do not particularly understand the reason for use of the contrastive loss. Whilst it does appear to have performance benefits, that is not the primary purpose of this paper - the purpose and narrative revolves around set-based prompting instead. A like-for-like comparison should eschew any change in loss function.
6. The results on perplexity in Table 1 are also mystifying. I do not understand - and the authors attempt to provide no explanation for - the significant decrease in language modelling perplexity after their fine-tuning procedure for Llama-2-7B-Chat.
The results demonstrated on the summarisation task with the use of set-based-finetuning are poor. Although, I do appreciate the inclusion of this result - and the thought behind this experiment.
7. Figure 2 and Figure 4 are very poorly presented in my opinion - it is not at all clear what the box plots and black dots refer to (although I was able to grok it eventually), and the use of ‘treatment’ and ‘control’ as terms is unnecessarily obtuse.

I encourage the authors to focus more narrowly on the core message of the paper - finetuning with a set-based methodology - and expand on the direction around its use for tasks beyond MCQ. Fundamentally, however, an explanation must be provided for why even a standard baseline finetuning methodology results in such poor task performance.

---

### Official Review · Reviewer_EXEG · 2025-03-02
**This paper addresses the critical issue of order dependence in LLMs for multiple-choice QA tasks.**

**Rating:** 6
**Confidence:** 3

**Review:**

### **Summary**

This paper addresses the critical issue of order dependence in LLMs for multiple-choice QA tasks. The authors propose fine-tuning models with Set-Based Prompting (SBP) and a margin-based contrastive loss, demonstrating improved robustness to answer permutations while preserving general language capabilities. The approach is well-motivated, methodologically sound, and empirically validated, though its applicability beyond QA tasks remains unproven.

### **Strengths**
1. **Practical Focus**: Addresses a known issue (order dependence) in LLMs, relevant for QA reliability.
2. **Empirical Validation**: Demonstrates improved robustness to permutations on MMLU, CSQA, and ARC benchmarks.
3. **Parameter Efficiency**: Uses LoRA for fine-tuning, reducing computational costs.

### **Weaknesses**
1. **Incremental Contribution**: Integrates existing techniques (SBP + contrastive loss) without novel algorithmic innovation. Merely fine-tuning on SBP-formatted data is a straightforward solution to prior SBP limitations.
2. **Narrow Scope**: Fails to generalize beyond QA tasks (e.g., summarization performance degrades under SBP). No exploration of tasks like ranking or dialogue.
3. **Technical Superficiality**:
   - SBP implementation details (e.g., attention masking, positional embeddings) are glossed over, hindering reproducibility.
   - Fixed margin (1.0) in the contrastive loss lacks justification; no ablation on margin sensitivity.
4. **Limited Data Diversity**: Fine-tuned exclusively on MMLU, raising doubts about cross-domain robustness.
5. **Overstated Significance**: Improvements are confined to specific QA setups, offering no broader insights into LLM invariance or bias mitigation.

---

### Official Review · Reviewer_4Ri4 · 2025-03-03
**Limited novelty, but a neat paper with strong trends in the results**

**Rating:** 6
**Confidence:** 3

**Review:**

### Summary:

SBP removes unwanted order dependence in MCQs. However, SBP is not in the training data distribution. This paper therefore investigates finetuning so that SBP can be used without the format being out of distribution for the LLM. They investigate both normal CE and also a contrastive loss, and find that the contrastive loss works significantly better.

### Strengths:

- Clearly written.
- Carefully constructed experiments and strong trends in the results.
- I appreciated the 4.6 Summarization Task paragraph. There should be more encouragement for papers to explain things that did not work, as there is often a lot that can be learned here.

### Weaknesses:

- Limited novelty, but in general a very neat paper.
- Why is figure 1 labelled as being specific to llama7b-base? Is the figure not illustrating the general principle of SBP?
- Line 86-7: Figure 1 doesn’t really visualize what is stated (“as visualized in Figure 1, SBP applies (1) modified attention masks that do not enforce strict left-to-right order within certain sub-sequences, and (2) identical or parallel positional embeddings for tokens in that sub-sequence.”)
- The paper would benefit from testing on another family of models and checking whether the same trends hold.
- What is meant by “modified accuracy”?

---

### Decision · Program_Chairs · 2025-03-04

**Decision:**

Reject

**Comment:**

The novelty of this work is very limited